# Tyrosine Kinase Inhibitor Profiling Using Multiple Forskolin-Responsive Reporter Cells

**DOI:** 10.3390/ijms241813863

**Published:** 2023-09-08

**Authors:** Yamato Kasahara, Sakura Tamamura, Gen Hiyama, Motoki Takagi, Kazuya Nakamichi, Yuta Doi, Kentaro Semba, Shinya Watanabe, Kosuke Ishikawa

**Affiliations:** 1Department of Life Science and Medical Bioscience, Waseda University, 2-2 Wakamatsu-cho, Shinjuku-ku, Tokyo 162-8480, Japan; y.k4309@akane.waseda.jp (Y.K.); k_nakamichi@fuji.waseda.jp (K.N.); yu-ta0418do-it@moegi.waseda.jp (Y.D.); ksemba@waseda.jp (K.S.); 2Japan Biological Informatics Consortium (JBiC), 2-45 Aomi, Koto-ku, Tokyo 135-8073, Japan; tamamura@jbic.or.jp; 3Translational Research Center, Fukushima Medical University, 1 Hikarigaoka, Fukushima 960-1295, Japan; hiyamag@fmu.ac.jp (G.H.); motokitakagi02@gmail.com (M.T.); swata@mvc.biglobe.ne.jp (S.W.)

**Keywords:** trap vector, reporter cell, panel analysis, forskolin, dasatinib, tyrosine kinase inhibitor

## Abstract

We have developed a highly sensitive promoter trap vector system using transposons to generate reporter cells with high efficiency. Using an EGFP/luciferase reporter cell clone responsive to forskolin, which is thought to activate adenylate cyclase, isolated from human chronic myelogenous leukemia cell line K562, we found several compounds unexpectedly caused reporter responses. These included tyrosine kinase inhibitors such as dasatinib and cerdulatinib, which were seemingly unrelated to the forskolin-reactive pathway. To investigate whether any other clones of forskolin-responsive cells would show the same response, nine additional forskolin-responsive clones, each with a unique integration site, were generated and quantitatively evaluated by luciferase assay. The results showed that each clone represented different response patterns to the reactive compounds. Also, it became clear that each of the reactive compounds could be profiled as a unique pattern by the 10 reporter clones. When other TKIs, mainly bcr-abl inhibitors, were evaluated using a more focused set of five reporter clones, they also showed unique profiling. Among them, dasatinib and bosutinib, and imatinib and bafetinib showed homologous profiling. The tyrosine kinase inhibitors mentioned above are approved as anticancer agents, and the system could be used for similarity evaluation, efficacy prediction, etc., in the development of new anticancer agents.

## 1. Introduction

We previously used transposon-mediated genomic manipulation to develop a promoter trap vector system for the efficient isolation of cells that can express reporter genes in response to a specific condition [1,2]. This promoter trap system efficiently produces reporter cells expressing enhanced green fluorescent protein (EGFP) and firefly-luciferase (Fluc) as reporter genes in response to arbitrary conditions and is sufficiently sensitive to amplify and detect changes of the promoter activity in response to a reactant, even with limited endogenous gene induction [1,3]. Using this system, we produced reporter cells that respond to the expression of the *c-Myc* gene [1], endoplasmic reticulum (ER) stresses [1], glucocorticoids [3], or vitamin A/D [4]. Although identifying the responding gene in the reporter cell is not always necessary, once identified, an unexpected gene is occasionally found as a novel marker. For example, *OSBPL9* was identified as a novel ER-stress-responsive gene but was overlooked in public data because of the low score [1]. Thus, almost random screening is highly attractive in identifying marker genes without being bound by past data and models and without narrowing down multiple candidates. Even if the response genes cannot be identified, they can be used as a tool for the quantitative evaluation of cellular responsiveness to biologically active substances.

We previously noticed that reactivity to vitamin A and its analogs varied between two reporter cells [4]. A characteristic profile of each compound was obtained as a unique pattern of dose-response curves, proposing that a more detailed pattern could be obtained by preparing additional reporter cells, and a bioanalyzer system could be constructed. Herein, we report that forskolin-responsive cells were found to be broadly responsive to multiple compounds, including the tyrosine kinase inhibitors (TKIs) dasatinib and cerdulatinib, and that unique response patterns for each compound were obtained. This strategy and findings may be a prototype for future research for drug evaluation and development.

## 2. Results

### 2.1. Isolation of Multiple Forskolin Reporter Cell Clones

While generating and evaluating a series of reporter cells with our highly sensitive promoter trapping method [3] (Figure 1a) and with K562 as the parental strain, we found that in a compound library containing 1327 FDA-approved drugs, a forskolin-responsive reporter cell clone (named #F1) was reactive to compounds other than forskolin. Detailed analysis confirmed that the top reactive compounds (PMA, vincristine, dasatinib, and cerdulatinib) were indeed reactive (Appendix A). These compounds are seemingly unrelated to the pathway involving adenylate cyclase, a known target activated by forskolin. In this study, we consequently examined whether any clones of forskolin-responsive cells would respond similarly to these compounds, and we additionally generated forskolin-responsive cells and obtained nine new clones (#F2 to #F10, Figure 1b). Splinkerette PCR demonstrated that the genomic insertion sites of the trap vector were different among these clones (Appendix A).

### 2.2. Each Forskolin-Responsive Clone Has a Unique Reaction Pattern for Multiple Compounds

The reactivity of the 10 forskolin-responsive clones to PMA, vincristine, dasatinib, and cerdulatinib was quantitatively evaluated using the luciferase assay (Figure 2a). Hierarchical clustering analysis using the values from this experiment revealed that each clone exhibited a different reactive pattern and that each of these drugs can be uniquely profiled by the reactivities of the ten clones (Figure 2b).

To better visualize these differences, we plotted the value in Figure 2b on a radar chart with each compound as the axis (Appendix A, Figure 3). The graphic shapes depicted in this figure may reflect the regulation of reporter gene expression. Thus, the more similar the shapes are, the more likely it is that the expression controls are similar.

### 2.3. Compound Profiling Using Multiple Forskolin Reporter Cells

The same values in Figure 3 were plotted on a radar chart (Figure 4), using the axis as each clone (Appendix A). The plotted shapes in Figure 4 should reflect the specific properties of each compound, and therefore, the more similar they are, the closer the biological activity of the compounds.

### 2.4. Evaluation of Other Tyrosine Kinase Inhibitors

We then evaluated seven other arbitrary TKIs known to target at least BCR-ABL, and additionally several other tyrosine kinases, such as KIT, SRC, LYN, and JAK3 [7], and a dasatinib–precursor compound (here we call CAY10697) using four clones that exhibit reactivity to dasatinib (#F1, #F2, #F6, and #F9) and one that did not (#F4) (Figure 2a). Each clone was treated with 0.5, 1, 2, and 10 μM of each reagent for 18 h, and a luciferase assay was then performed to obtain fold change values relative to treatment without drug (0 μM) (Figure 5). From these results, it can be seen that the five reporter cells are differentially responsive to each compound. For the most basic profiling, we analyzed the condition at 4 μM for 18 hours (Figure 6a). This is the concentration and time at which all compounds induce a detectable expression of luciferase and do not kill cells. The heatmap from hierarchical clustering analysis, and a radar chart based on the average fold change values are shown to facilitate visual clarity (Figure 6b,c). These results showed that dasatinib and bosutinib form an isolated cluster; adaphostin is another isolated cluster; and the others forms a different cluster, of which imatinib and bafetinib are particularly similar. These similarities and differences are apparently not due to structural differences [8,9,10] and biochemical affinity to BCR-ABL [9]. Unique commonality between compounds with homologous patterns could not be found in the past literature, although a target profile by a hybrid drug–protein/protein–protein interaction network for nilotinib, dasatinib, bosutinib, and bafetinib [11] seems to be partially consistent with our result. Adaphostin, known as a drug for having kinetically different modes of action from that of imatinib (STI571) for leading to apoptosis [12,13], showed a strong unique reactivity to a single clone (#F4). These results suggest that it is possible to study the homology and differences in compound biological activities against TKIs using multiple forskolin-responsive reporter cells.

### 2.5. Identification of Forskolin-Responsive Genes by Splinkerette PCR or 5′Rapid Amplification of cDNA Ends (5′RACE)

To determine which genes the trap vectors were inserted into and affected under the control of expression, we analyzed the vector insertion site of the genome by sequencing the DNA band of the splinkerette PCR (Appendix A) or the transcripts fused to the reporter gene (*GAL4FF*) by 5′RACE (Appendix A). As a result, we succeeded in identifying response genes (Table 1). *MALRD1* (also known as *DIET1*) encodes a conserved protein characterized by multiple MAM (Meplin-A5-protein tyrosine phosphatase μ) and LDLR A2 (low-density lipoprotein receptor A2) domains [14]. *CXCL3* gene encodes a member of the CXC subfamily of chemokines, and this protein is a secreted growth factor that signals through the G protein-coupled receptor, CXC receptor 2 [15,16]. CXCL3 is an autocrine ligand for CXCR2, and the signaling pathway originating from CXCR2 activates PPARγ to activate *CXCL3* transcription [17]. Since PPARγ expression is activated by forskolin stimulation [18], *CXCL3* could be induced by forskolin stimulation. *LGALS1* gene encodes Gal1, one of 15 members of a family of β-galactoside-binding proteins called galectins, which are described to function as homodimeric proteins consisting of 14.5 kDa subunits. Gal1 expression is regulated via the luteinizing hormone/choriogonadotropin receptor (LHCGR), a common receptor for LH and hCG [19]. Interestingly, hCG stimulation in hGL5 cells stimulated intracellular accumulation of cAMP in hGL5 cells [20]. So, if some feedback exists, there may be a common mechanism for responding to forskolin.

To examine whether the endogenous genes also respond to forskolin in the same manner as a reporter, we evaluated their mRNA expression by real-time PCR. The expression of all identified genes was upregulated by the addition of forskolin (Figure 7). Since the fold change is less than twofold for any of the identified genes, the amplification mechanism by the GAL4-UAS system in the reporter cells seem to be essential for detection of response and reliable quantitative analyses.

## 3. Discussion

Herein, we showed that TKIs could be characteristically profiled by using multiple forskolin reporter cells. The inspiration for this study was the observation that reporter cells obtained by forskolin responded not only to forskolin but also to several other drugs. By contrast, the reporter cells we previously generated had mostly been specific and unresponsive to many other drugs tested; forskolin-responsive cells were rare special cases that respond to seemingly unrelated compounds. In our previous study, cells that responded to calcitriol, an activated form of vitamin D, also responded strongly to the vitamin A compounds, bexarotene, 9-cisRA, tretinoin, and isotretinoin [4]. This was also considered as a rare case where reporter cell clones showed reactivity to both the stimulus for cloning and to other compounds. We previously discussed that vitamin A variants can be classified by multiple reporter cells. The key to profiling compounds with multiple reporter cells is to find exceptional compound responsiveness, as we successfully experienced with the forskolin- and calcitriol-responsive cells. Reporter cells that respond specifically to the stimulus used for their creation cannot be used for profiling various compounds. Therefore, it is necessary to comprehensively investigate the responsiveness to many compounds without preference. At present, we do not know whether the same profiling analysis is possible for any other compounds and whether it was fortuitous that TKIs could be evaluated. We hope to expand the number of reporter cells and test compounds and find a set of reporter cells that can evaluate different compounds.

It is currently unclear why forskolin-responding cells also respond to TKIs. This is interesting because these compounds are not apparently linked by identified pathways. Conceivably, various feedback pathways may be activated, including the autocrine mechanism, or multiple regulatory control sequences may be present in the promoter region of the responding gene. For the calcitriol-responsive cells mentioned above, the *TSKU* gene was identified as a responsive gene that had multiple RARE and VDRE sequences where nuclear receptors for vitamins A and D, respectively, can bind. This may be the reason why multiple vitamin compounds react in these cells. Conversely, dasatinib has not been reported to bind to nuclear receptors but is a potent inhibitor of BCR-ABL and is a standard drug of choice for molecular targeted therapy for malignant tumors. Reporter cells were significantly less responsive to imatinib and nilotinib (Figure 6c), which are standard BCR-ABL inhibitors, suggesting that a pathway different from the BCR-ABL pathway is involved in responsiveness. Dasatinib is not a selective BCR-ABL inhibitor but a multi-tyrosine kinase inhibitor, e.g., EGFR, EPHA2, EPHB1, BTK, MAP3K4, MAP3K14, DDR1, GAK, etc. [21,22]. Similarly, imatinib and nilotinib also target proteins beyond BCR-ABL, each with specific differences compared to dasatinib. Notably, they are known to commonly bind and inhibit the non-kinase target NQO2 [23]. Thus, it is likely that there are multiple pathways in common with the forskolin response pathway that are different from BCR-ABL pathway. When trying to directly generate dasatinib- or imatinib-responsive cells, reporter cells could not be generated without the prior generation of dasatinib- or imatinib-resistant cells. Our procedure for generating reporter cells is disadvantaged by requiring that clones cannot be selected unless they are stimulated at least once, and therefore the cells must survive the stimulation and reversibly return to a steady state. If the cells are addicted to BCR-ABL, they will probably not survive in the presence of its inhibitors unless the stimulation and concentration conditions are appropriate. Even if dasatinib- or imatinib-resistant cells are created, the cells may not respond to them depending on the resistance mechanism involved. However, this study found that another stimulus can be substituted when producing dasatinib- or imatinib-responsive cells, providing a way around this problem. Comprehensive analysis could discover cells that report a drug that inevitably cannot produce reporter cells. Therefore, it is highly valuable to continue to generate and pool randomly obtained reporter cells that are unconstrained by existing models. As promoter trapping is almost random (although some bias toward transcriptionally active regions has been recognized [24]), it would be better if the stimuli for producing reporter cells were also random. Furthermore, if the goal is to obtain multiple reporter cells, no matter what the reaction is, a single stimulus to produce responsive cells is not needed. For example, a mixture of serum, culture supernatant (conditioned medium), or other biological liquid samples could be used as a stimulant. This conceptual methodology has the potential to save a significant amount of time and cost compared with carefully handling each single stimulus, and can find undiscovered bioactive substances.

The luciferase assay data were obtained with high sensitivity and accuracy. It would be impossible to obtain data from microarrays with the same level of detection. In real-time PCR analysis, the fold change value for forskolin-induced mRNA level was smaller than 2 for every responsive endogenous gene (Figure 7). Reporter cells obtained by the promoter trap method express the reporter gene via the GAL4-UAS system [5,6,25] and can thus respond to stimuli with extreme sensitivity. This is the greatest advantage of this evaluation system. Since a cellular organism is used as a detector, this can be termed as a highly sensitive bioanalyzer system. For simplicity, we limited the profiling to a single concentration condition, but we believe that dose analysis will further deepen our understanding of the drug’s properties. In fact, analysis of the 5 response clones at 2 concentrations of dasatinib showed difference in response depending on the dose, especially for #F2 (Appendix A). Although new ideas for data processing still need to be developed, the use of such dose dependency for profiling in the future will make it an even more reliable analytical tool.

## 4. Materials and Methods

### 4.1. Cell Culture

Human chronic myelogenous leukemia K562 cells (JCRB0019) were maintained in RPMI medium (Fujifilm Wako, Tokyo, Japan) supplemented with 10% fetal bovine serum (Nichirei Biosciences, Tokyo, Japan), 100 μg/mL streptomycin sulfate, and 100 U/mL penicillin G potassium. Cells were incubated at 37 °C with 5% CO_2_.

### 4.2. Establishment of Reporter Cells Responsive to Forskolin

K562 cells were collected, and 1 × 10^6^ cells were co-transfected with 10 μg mixture of pCMV-hyPBase [26] and transposon donor vectors designed to trap the promoter activity of endogenous genes (Figure 1a) [3] at an OD_260_ ratio of 1:3 in 100 μL of Opti-MEM (Gibco, Billings, MT, USA). Electroporation was performed in a cuvette with 2 mm gap between electrodes using a NEPA21 electroporator (NEPAGENE, Chiba, Japan) at the following condition: poring pulse, 275 V; pulse width, 1 ms; pulse interval, 50 ms; repeat, 2; attenuation ratio, 10%; polarity, +; transfer pulse, 20 V; pulse width, 50 ms; pulse interval, 50 ms; repeat, 5; attenuation ratio, 40%; polarity, +/−. Immediately after electroporation, all the cells were suspended in media, centrifuged, resuspended, and then spread onto a 12-well plate in 1 mL culture medium. After cell propagation, the EGFP-positive population (considered to express EGFP constitutively) was removed by three to four rounds of cell sorting using SH800Z cell sorter (Sony, Tokyo, Japan). Sorted cells were then propagated on a 10 cm dish scale and were stimulated with 10 μM forskolin overnight at 37 °C in a CO_2_ incubator; subsequently, high EGFP-expressing cells were single-cell sorted into 96-well plates. After the propagation of these cells, their reagent responsiveness was checked under a fluorescence microscope. Whenever multiple clones were obtained from the same parental cells, the genomes were assessed using splinkerette PCR analysis [1,27,28] to select independent clones as assumed from their band patterns.

### 4.3. Luciferase Assay

The luciferase assay was performed as previously described by Siebring-van Olst using a slightly modified version [29]. Briefly, cells were spread on a white-colored 96-well cell culture plate. Eighteen hours after the addition of the stimulating reagent (to a total volume of 90 μL), the plate was set on a plate reader (Berthold Technologies Japan, Tokyo, Japan, TriStar2S LB942) and the following program was executed: 90 μL of 2 × FLAR (40 mM Tricine [pH 8], 200 μM EDTA, 2.2 mM MgCO_3_, 5.3 mM MgSO_4_∙7H_2_O, 500 μM ATP, 2% TritonX-100, 40 mM DTT, and 500 μM D-luciferin) was dispensed to all wells of the plate, which was then shaken for 180 s, followed by photon counting for 3 s per well after a 0.2 s delay. The means of at least three independent experiments (in each independent experiment, three samples were measured for one condition) were graphed using GraphPad Prism software ver.6.07. FDA-approved drug library that includes 1325 drugs we tested was purchased from Selleck Biotech (Kanagawa, Japan, cat#L2000). A list of reagents is shown in Table 2.

### 4.4. Hierarchical Clustering Analysis

For hierarchical clustering computing, the Euclidean distance following Ward’s method was used with R (ver. 4.2.2) [30]. Heatmaps were depicted by pheatmap R package (ver. 1.0.12) [31]. The binary logarithm was calculated after adding 1 to the mean fold change values from the luciferase assay in Figure 2a or Figure 6a to better clarify the difference. Radar charts were drawn using Excel software (2021) with the same values as the hierarchical clustering.

### 4.5. 5′RACE

cDNA was synthesized from 5 μg of total RNA extracted from each clone by ISOGEN reagent (Nippon Gene, Tokyo, Japan) using SuperScript III Reverse Transcriptase (Thermo Fisher Scientific, Waltham, MA, USA) according to the manufacturer’s protocol, using the primer 5′-TGGCCAGTCTATCAGTAAC-3′ (named GSP1) that anneals specifically to the transcription factor GAL4 derived from the trap vector. Then, 2 U of RNaseH and 1 U of RNaseA were added to the reverse transcription solution and reacted at 37 °C for 2 h to degrade any residual RNA. The cDNA was purified using the QIAEX^®^II Gel Extraction Kit (Qiagen, Valencia, CA, USA) and eluted with 25 μL of water. To add poly-C oligonucleotides to the 5′-end, the 5 μL of eluted cDNA was reacted in a tailing buffer made up of 10 mM Tris-HCl (pH 8.4), 25 mM KCl, 1.5 mM MgCl_2_, 0.2 mM dCTP, and 7 U of terminal deoxynucleotidyl transferase (Takara, Kyoto, Japan) at 37 °C for 10 min and then heat inactivated at 65 °C for 10 min. Nested PCR was then conducted to amplify the specific GAL4 fusion gene sequence, using KOD FX Neo (Toyobo, Osaka, Japan) (Table 3) and the primers listed in Table 4. The amplified PCR fragments (Appendix A) were collected after 1.5% agarose gel electrophoresis, purified with the QIAEX^®^II Gel Extraction Kit, and ligated to the PCR-amplified pBluescript II SK(−) fragment using the In-Fusion^®^ Cloning Kit (Takara). The fused plasmid was then introduced into *Escherichia coli* (DH5α) cells and incubated on an ampicillin-containing LB plate at 37 °C overnight; then colony PCR was conducted using primers presented in Table 4. After 2% agarose gel electrophoresis, if the DNA bands (Appendix A, right) appeared, colonies of the band’s origin were picked up and cultured in ampicillin-containing YT liquid microbial growth medium at 37 °C overnight, and the fused plasmids were purified with the Fastgene Plasmid Mini Kit (NIPPON Genetics, Tokyo, Japan), being analyzed their sequence using the oligonucleotide primer 5′-GGACTGTACCTACACTCCCAATTG-3′. The obtained sequence was analyzed with the BLAT tool of the UCSC genome browser (https://genome.ucsc.edu/cgi-bin/hgBlat accessed on 23 August 2023), and the responsive gene candidates were identified (Appendix A and Table 1).

### 4.6. Real-Time PCR

RNA was extracted from the cultured cells using ISOGEN reagent (Nippon Gene, Tokyo, Japan). Reverse transcription was performed using the SuperScript III First-Strand Synthesis Kit (Thermo Fisher Scientific) with oligo-dT primer as the primer mix. The THUNDERBIRD SYBR qPCR Mix (TOYOBO, Osaka, Japan) was used for the PCR, which was executed on the StepOnePlus Real-Time PCR System (Applied Biosystems, Foster City, CA, USA). The primers used are listed in Table 5. The human *ACTB* gene was used as the internal control.

## 5. Conclusions

Quantitative evaluation using multiple forskolin-responsive cells revealed that the tyrosine kinase inhibitors exhibited unique patterns. This evaluation system using multiple reporter cells provides a new quantitative method using cellular biological responses and may be a useful tool for pharmaceutical development, including anti-cancer drugs.

## Figures and Tables

**Figure 1 ijms-24-13863-f001:**
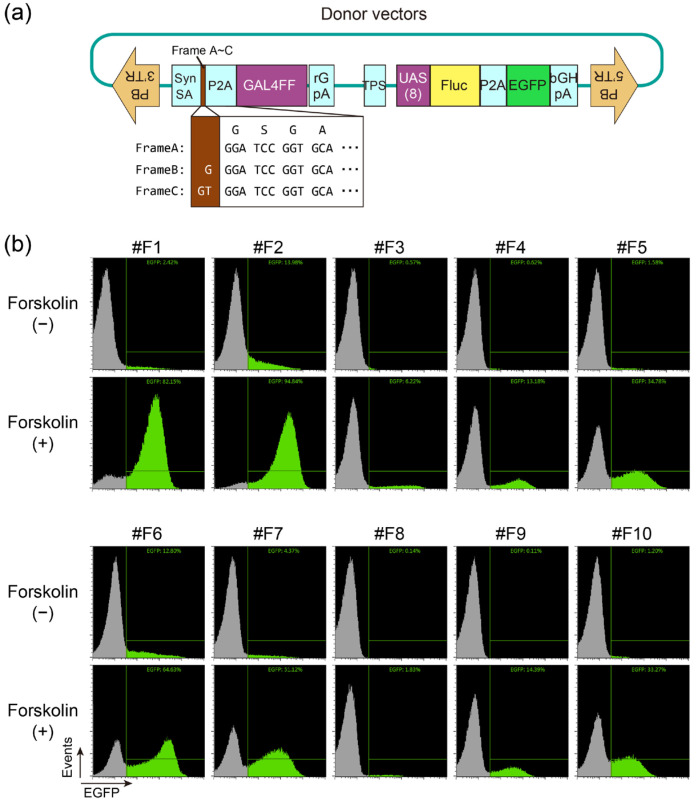
(**a**) The donor vector structures of the transposon-mediated promoter-trapping system. TR, terminal repeat; Syn SA, synthetic splicing acceptor; P2A, 2A peptide derived from porcine teschovirus-1; GAL4FF, an extremely trimmed minimal DNA-binding site of the yeast GAL4 transcription factor with two repeats of the minimal transcription activation module from VP16 [5,6]; rGpA, rabbit beta-globin polyadenylation signal; TPS, transcription pause site; bGHpA, bovine growth hormone gene polyadenylation signal. Numbers in parentheses represent the number of repeats. Please refer to [3] for further details. (**b**) Generation of a series of different forskolin-responsive reporter cell lines. Promoter trap vectors (**a**) were introduced into K562 as described in Material and Methods, and forskolin-responsive cells expressing Fluc-P2A-EGFP reporter were cloned. Each clone was treated overnight without (−) or with (+) 10 μM forskolin and quantitatively evaluated using flow cytometry. EGFP-positive cells are shown in green.

**Figure 2 ijms-24-13863-f002:**
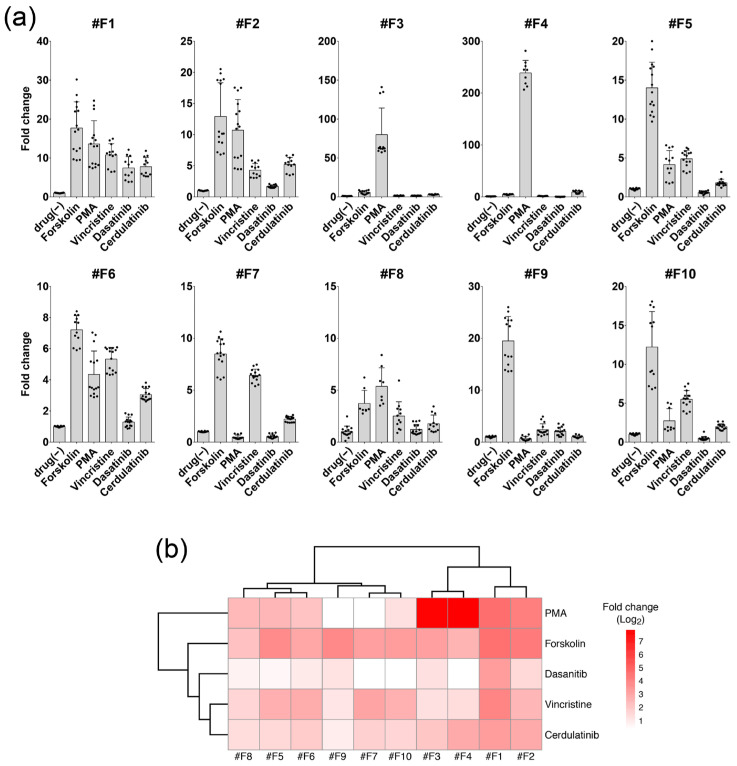
(**a**) Quantitative evaluation of drug responsiveness of each forskolin-responsive clone via luciferase assay. Cells were treated with the indicated reagents overnight, and the luciferase assay was performed. To show the simplest profiling, only the result of 0.5 μM concentration, which has sufficient detection of luciferase activity and does not kill cells with any of the reagents shown here, is presented. The graph shows the mean ± SD of 3–6 experiments (in each independent experiment, three samples were measured for one condition). Fold change values are indicated by the average value of drug (−) as 1. (**b**) Heatmaps obtained by the hierarchical clustering analysis based on the average fold change values of (**a**) are depicted.

**Figure 3 ijms-24-13863-f003:**
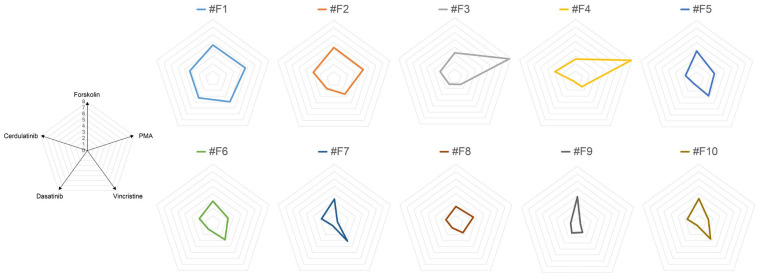
Radar chart display of the values from the luciferase assay in Figure 2. Here, the radar chart in Appendix A is displayed separately for each clone.

**Figure 4 ijms-24-13863-f004:**
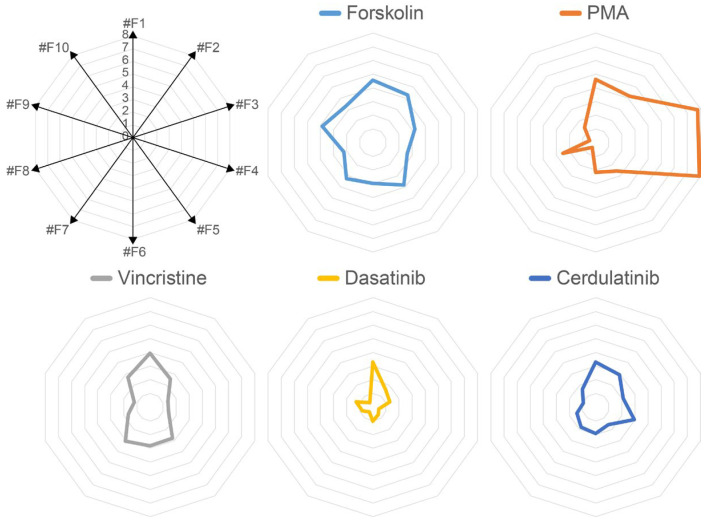
Radar chart display of the values from the luciferase assay in Figure 2 using the axis as each clone. Here, the radar chart in Appendix A is displayed separately for each reagent.

**Figure 5 ijms-24-13863-f005:**
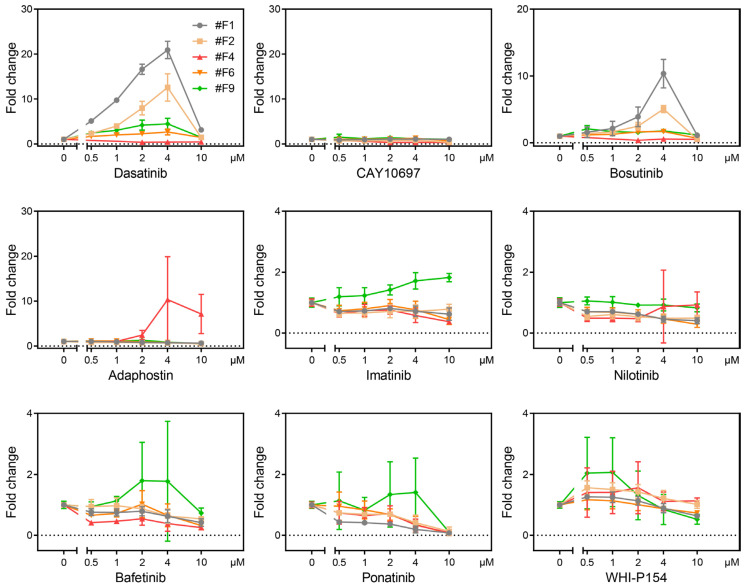
Dose-response analyses of reporter clones for TKIs. Each reagent was treated for 18 h at the indicated concentration of drugs, and then a luciferase assay was performed. The graph shows the mean ± SD (9 ≤ *n* ≤ 21) of the fold change value against drug (−) (three to six experiments were conducted, with three independent samples measured in each experiment).

**Figure 6 ijms-24-13863-f006:**
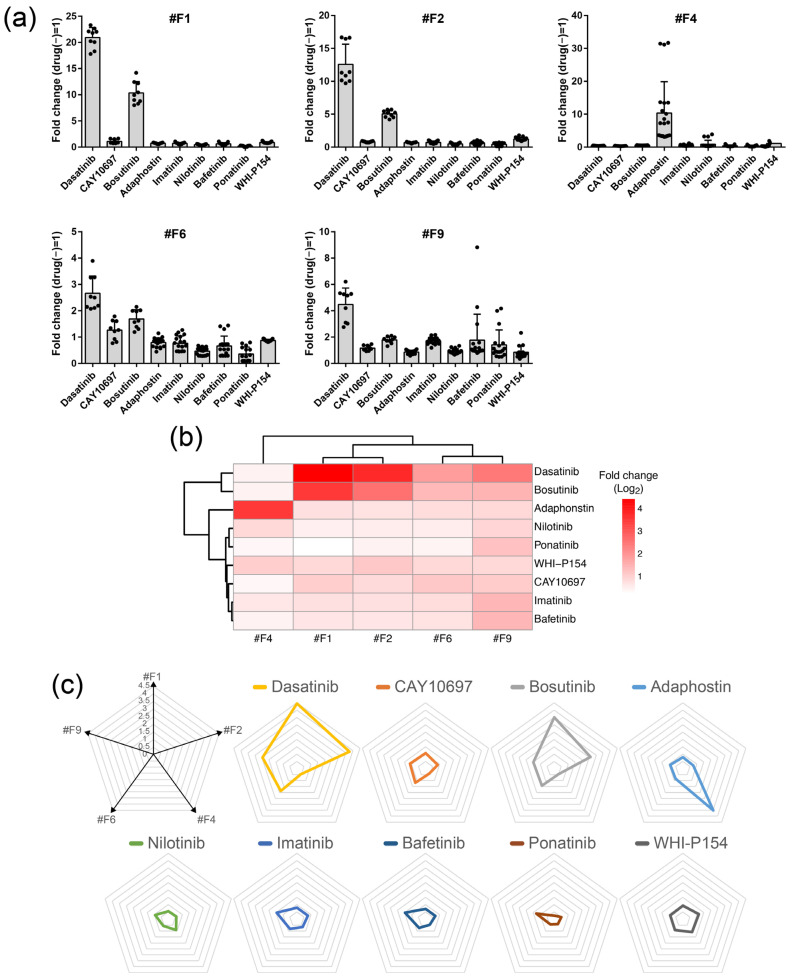
(**a**) Reactivity of five forskolin-responsive reporter cells to TKIs. For the simplest profiling analysis, the 4 μM results were extracted from the results in Figure 5 and redrawn as a bar graph for each reporter clone. The graph shows the mean ± SD (9 ≤ *n* ≤ 21) of the fold change value against drug (−) (three to six experiments were conducted, with three independent samples measured in each experiment). (**b**) Hierarchical clustering analysis was performed based on the fold change values from (**a**). (**c**) Radar chart display of the value in (**b**) plotted on each clone axis. Here, the radar chart in Appendix A is displayed separately for each reagent.

**Figure 7 ijms-24-13863-f007:**
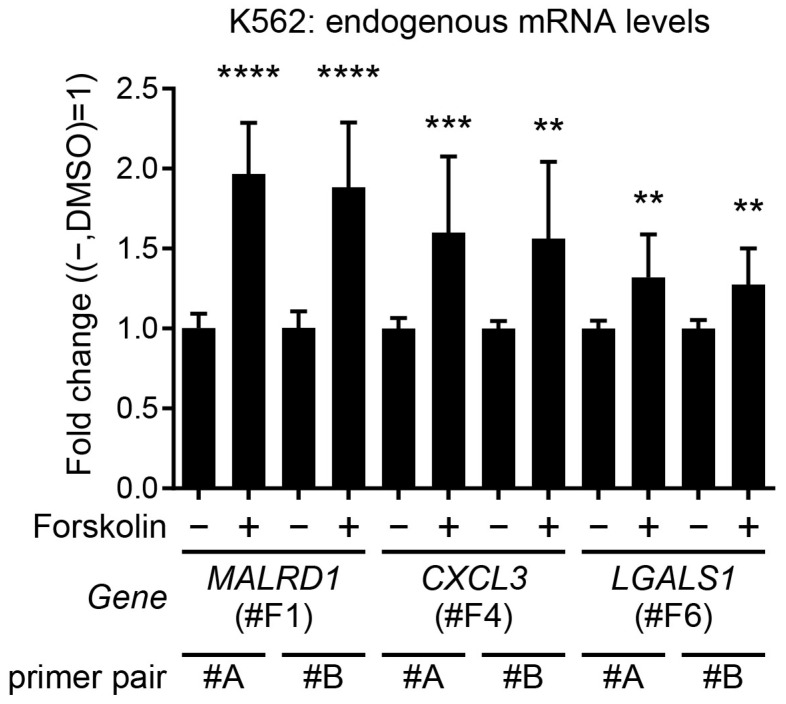
Real-time PCR analysis for endogenous expression of genes identified as forskolin. K562 cells without vector transfection were treated overnight without (−) or with (+) 10 μM forskolin. RNA was extracted and reverse-transcribed, and then analyzed by real-time PCR. Each gene was analyzed with two different primer pairs. *ACTB* was used as an internal control. Clone numbers in parentheses indicate that the gene was identified by the clone. The graph shows the mean ± SD (*n* = 10) of the fold change value against drug (−) (three independent experiments were conducted, with three or four independent samples measured in each experiment). Statistically, *t*-test was performed. ** *p* < 0.01, *** *p* < 0.001, **** *p* < 0.0001.

**Table 1 ijms-24-13863-t001:** Possible responsive genes as revealed by splinkerette PCR or 5′RACE.

Clone#	Frame	^1^ Predicted Response Gene	Method
#F1	A	*MALRD1*	Splinkerette PCR
#F2	A	^2^ N.D.	-
#F4	B	*CXCL3*	5′RACE
#F6	B	*LGALS1*	5′RACE
#F9	C	N.D.	-

^1^ It cannot be denied that there may be other trapped genes that could not be identified by these methods. Also, the possibility of trapping more than one gene promoter cannot be ruled out. ^2^ N.D., not determined; that is, it could not be identified by the analyses.

**Table 2 ijms-24-13863-t002:** Reagents used in this study.

Reagent	Manufacturer	Product Number	CAS Number
Forskolin	Wako, Saitama, Japan	063-02193	66575-29-9
PMA (Phorbol 12-myristate 13-acetate)	Cayman Chemical, Ann Arbor, MI, USA	10008014	16561-29-8
Vincristine Sulfate	Wako	220-02301	2068-78-2
Dasatinib	Wako	ST-7591	302962-49-8
Cerdulatinib	Wako	20076	1198300-79-6
Bosutinib	Selleck	S1014	380843-75-4
CAY10697	Cayman Chemical	CAY10697	302964-08-5
Adaphostin	Sigma-Aldrich, St. Louis, MO, USA	SML1110	241127-58-2
Imatinib	Cayman Chemical	13139	152459-95-5
Nilotinib	Cayman Chemical	10010422	641571-10-0
Bafetinib	Selleck	S1369	859212-16-1
Ponatinib	Selleck	S1490	943319-70-8
WHI-P154	Selleck	S2867	211555-04-3

**Table 3 ijms-24-13863-t003:** Conditions used for the 5′RACE PCR (common to 1st and 2nd nested PCR).

Temperature	Time	Repeat
94 °C	2 min	1
98 °C	10 s	30
60 °C	30 s	
68 °C	2 min	
68 °C	5 min	1
15 °C	∞	-

**Table 4 ijms-24-13863-t004:** PCR primers for identifying responding genes by 5′RACE.

Purpose	Primer Sequence (5′ to 3′)
Reverse transcription	TGGCCAGTCTATCAGTAAC (GSP1)
5′RACE 1st PCR	GGCCACGCGTCGACTAGTACGGGGGGGGGGGGGGGG (AAP)
	AGCTTTGATGTCTTGCAGGGAATC (GSP2)
5′RACE 2nd PCR	GGCCACGCGTCGACTAGTAC (AUAP)
	GTTGTTCAAGCCTCTCCAGTCTTG (GSP3)
Amplifying pBluesciptII SK(−) for subcloning	AGAGGCTTGAACAACCTTATCGATACCGTCGACCTCG
	AGTCGACGCGTGGCCCTTGATATCGAATTCCTGCAGC
Colony PCR for subcloned plasmid	GGAAACAGCTATGACCATG
	GGACTGTACCTACACTCCCAATTG

**Table 5 ijms-24-13863-t005:** PCR primers for the real-time PCR.

Gene, Primer Pair#	Primer Sequence (5′ to 3′)
*MALRD1*, #A	CTCACATCACAGTTGCAGTCTTG
	GACAAAGGCACAGTTACCACTTC
*MALRD1*, #B	ACAGCCTGCAGTCTTACTCAAG
	TGCTGGGAAGGATTTGGAAGTAG
*CXCL3*, #A	GAAGTCATAGCCACACTCAAGAATG
	CGCTGATAAGCTTCTTACTTCTCTC
*CXCL3*, #B	GTCATAGCCACACTCAAGAATG
	GCTTCTTACTTCTCTCCTGTCAG
*LGALS1*, #A	CTGGACTCAATCATGGCTTGTG
	TTGCTGTCTTTGCCCAGGTTC
*LGALS1*, #B	CTGGGCAAAGACAGCAACAAC
	ACACCTCTGCAACACTTCCAG
*ACTB*	GGCGGCAACACCATGTACCCT
	AGGGGCCGGACTCGTCATACT

## Data Availability

The data presented in this study are available on request from the corresponding author.

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
