# Peer review of "Tyrosine Kinase Inhibitor Profiling Using Multiple Forskolin-Responsive Reporter Cells"

_ijms, 2023, doi:10.3390/ijms241813863_

Round 1

Reviewer 1 Report

One major concern about the manuscript is the drug testing in the luciferase assay. The authors only tested drugs at a single concentration (0.5 μM) for one time point (overnight) to assess the drug responsiveness of each forskolin-responsive clone. While this approach allowed for a preliminary evaluation of drug activity, it lacks a clear rationale for why this specific condition was chosen for all the drugs. It must be noted that changes in concentrations and treatment durations can significantly influence the pattern of drug response in the luciferase assay. Thus a single time point and concentration may not be representative of the overall drug behavior.

Furthermore, each drug may have distinct mechanisms and physicochemical properties, even if they are all classified as tyrosine kinase inhibitors. Simply comparing the response at one fixed concentration does not provide a reliable understanding of the compound's behavior.

Additionally, differences in fold change do not necessarily reflect the true regulation of reporter gene expression, as compounds can differ in the plateaus of their dose-response curves. In the manuscript, the radar chart display based solely on one condition can be misleading.

In light of these concerns about the validity of their methodology, it is crucial for the authors to perform a dose-response analysis for all the compounds in the luciferase assay. Only then can they conduct meaningful compound profiling, make more accurate comparisons among the drugs’ responses, and draw meaningful conclusions.

Reviewer 2 Report

The manuscript by Kasahara et al. shows the new cellular tool that can initially evaluate new tyrosine kinase inhibitors. Though the paper itself is well written it suffers from a lack of clarity which is why it requires substantial revision.

1.      First, for me this is not a regular article, thus it should be changed to short communication.

2.      In the introduction and materials and methods sections the authors must describe the system in more detail so that it can be understood by the non-specialist. They should describe which vectors carry EGFP and luciferase reporter genes.

3.      Figure 2. The authors used 0.5µM of compounds to treat cells and for Figure 5 they used 4µM concentrations. Why? The authors should explain these differences particularly that different concentrations produced different patterns e.g. dasatinib (Figure 4 and Figure 5C).

4.      Discussion. Lines 167-171. Dasatinib is not a selective BCR-ABL inhibitor but a multi-tyrosine inhibitor e.g. EGFR, EPHA2, EPHB1, BTK, MAP3K4, MAP3K14, DDR1, GAK, etc. Thus the authors should cite Cancers (Basel). 2019 11(5):673, Nat. Chem. Biol. 2010, 6, 291-299. Similarly for imatinib and nilotinib please cite Blood 2007, 110, 4055-4063.

5.      The authors mentioned (introduction section) that previously using a similar system they identified OSBPL9 as a new ER-stress responsive gene. Did they find any genes that are responsive to TKI used in the present study?

I am not native speaker thus for me the lanaguage is fine.

Round 2

Reviewer 1 Report

The authors have made significant efforts to address the raised concerns in the revised manuscript. Though the trends are unclear, the inclusion of the dose-response results will be valuable and informative to readers. I have no additional comments. The revised manuscript is now in a suitable state for publication.

Reviewer 2 Report

The authors improved the manuscript thus I have no further comments.

I am not native speaker and for me the language is fine.